# Indigenous Oral Health Inequalities at an International Level: A Commentary

**DOI:** 10.3390/ijerph17113958

**Published:** 2020-06-03

**Authors:** Lisa Jamieson, Dandara Haag, Helena Schuch, Kostas Kapellas, Rui Arantes, W. Murray Thomson

**Affiliations:** 1Australian Research Centre for Population Oral Health, School of Dentistry, University of Adelaide, Adelaide 5005, Australia; dandara.haag@adelaide.edu.au (D.H.); helenasschuch@gmail.com (H.S.); kostas.kapellas@adelaide.edu.au (K.K.); 2Fundacao Oswaldo Cruz Mato Grosso do Sul, Campo Grande 79074-460, Brazil; rui.arantes@fiocruz.br; 3School of Dentistry, University of Otago, Dunedin 9054, New Zealand; murray.thomson@otago.ac.nz

**Keywords:** indigenous, oral health inequalities, Shiffman and Smith’s Political Power Framework, social determinants of health

## Abstract

Oral health inequalities reflect social injustice. This is because oral health simultaneously reflects material circumstances, access to health services and inequities across the life course. Oral health inequalities between Indigenous and non-Indigenous populations are among the largest in the world. This paper provides a critical commentary on Indigenous oral health inequalities at an international level based on existing literature and policies. We include the role of systematic and institutionalized racism and how this enables the persistence and flourishing of Indigenous oral health inequalities. We discuss theoretical frameworks—including Shiffman and Smith’s Political Power Framework—that underpin the power constructs that contribute to those. This theory posits that power is exercised in four ways: (i) the power of ideas; (ii) the power of the issue; (iii) the power of the actors; and (iv) the power of the political context. We will demonstrate, using examples of Indigenous oral health inequalities from several countries, how intervening at key leverage points, acting simultaneously on multiple subsystems and counteracting the social determinants of health are crucial strategies for ameliorating Indigenous oral health inequalities at a global level.

## Commentary

The earliest empirical documentation of oral health inequalities occurred with the inception of research in dental public health [1]. Research in oral health since the 1950s (including from the World Health Organization) has consistently documented social inequalities [2,3,4]. Population-level approaches to reducing oral health inequalities have been implemented with varying degrees of success, the most notable being the use of fluoride in its various forms [5]. In many countries, including those with social policies that favor access to dental services, oral health inequalities appear to be increasing [6,7]. Indigenous inequalities in oral health are recognized as both persisting and flourishing at a global level [8]. For example, using estimates from National Oral Health Surveys in Australia, Canada and New Zealand, the prevalence of untreated dental caries was as follows: 49% among Aboriginal Australians but 23% among non-Aboriginal Australians; 35% among Canadian Aboriginals but 19% among non-Aboriginal Canadians; and 50% among New Zealand Maori but 34% among non-Maori New Zealanders. Pooled estimates from South American Indigenous peoples (from Brazil, Chile, Uruguay and Venezuela) show that the mean (d)ecayed, (m)issing and (f)illed (t)eeth (dmft) for Indigenous 5-year-olds was 5.7, the mean DMFT for Indigenous 12-year-olds was 3.1, the mean DMFT for Indigenous 15–19-year-olds was 5.5, the mean DMFT for Indigenous 35–44-year-olds was 19.4 and that for Indigenous 65–74-year-olds, it was 28.2. These estimates were far greater than those reported for non-Indigenous South American people.

The United Nations Indigenous Peoples’ Partnership defines being “Indigenous” as “having a historical continuity with pre-invasion and pre-colonial societies that developed on their territories, with Indigenous persons considering themselves as being distinct from other sectors of the societies now prevailing on those territories”. On an individual basis, an Indigenous person is one who belongs to an Indigenous population through self-identification and who is recognized and accepted by that population as one of its members. This preserves for these communities the sovereign right and power to decide who belongs to them, without external interference [9]. Globally, there are an estimated 476.6 million Indigenous people in nearly 90 countries. This represents 6.2 percent of the worldwide population.

Despite geographic and cultural differences observable at the international level, Indigenous persons share common problems related to the protection of their rights as distinct populations [10]. Indigenous peoples today are among the most disadvantaged and disenfranchised in the world [11]. This is because, in many countries, Indigenous groups have been subject to sustained government policies of colonization, discrimination and marginalization, along with strategic directives that focus on assimilation and, in some cases, cultural annihilation [12]. Such historical legacies have had marked impacts on Indigenous health, but it is only recently that some countries have established policies to address such disparities [13]. 

The social determinants of health provide a useful framework by which to conceptualize Indigenous health inequalities at a global level [14]. Indigenous societies are collectives of more than just “socially disadvantaged” persons, though. The poverty and inequality that they experience is a contemporary reflection of their historical treatment as peoples, which may not apply to other disenfranchised members of society. In Australia, this has been demonstrated by the Royal Commission into Aboriginal Deaths in Custody (which documented the links between social position and imprisonment [15]) and the National Inquiry into the Forcible Removal of Aboriginal and Torres Strait Islander Children from their Families, which reflected the inter-generational challenges for parenting, health, accessing care and protection of the removal of children during the assimilation period [16]. In New Zealand, an inquiry to address the high rates of removal of Māori children from their families by government agencies is currently being headed by key Maori leaders [17]. The current political context is very unfavorable for the 100 million Indigenous peoples in Brazil. The government has often disrespected the Indigenous constitutional rights that guarantee access to territory and health [18]. The federal government has been trying to approve laws projects that allow mineral exploration and energy generation in Indigenous lands. The country’s National Indian Foundation (FUNAI), which should be the official agency for protecting Indigenous territorial rights, has been weakened and politically controlled. In April 2020, FUNAI published a Normative Instruction that allows the occupation and sale of Indigenous lands that have not yet been officially regularized [19]. A recent inquiry in Canada has labelled as genocide the sustained and misreported killings of First Nation and Inuit Women over the last three decades, with 231 recommendations in the report [20]. These unique social determinants are disproportionately salient to Indigenous populations, and they need to be taken into account when considering specific outcomes of interest (such as oral health).

Systemic racism (also known as structural racism or institutional racism) has been defined as a covert and subtle form of racism expressed in the practice of social and political institutions [21,22]. It originates in the operation of established and respected societal forces, and thus receives less public condemnation than individual-level racism [23]. The 1999 Lawrence report (an inquiry into a racially motivated murder in the United Kingdom) defined systemic racism as “The collective failure of an organization to provide an appropriate and professional service to people because of their colour, culture, or ethnic origin. It can be seen or detected in processes, attitudes and behaviour which amount to discrimination through unwitting prejudice, ignorance, thoughtlessness and racist stereotyping which disadvantage minority ethnic people” [24]. Irrespective of country, systemic racism is reflected in inequities in wealth, income, criminal justice, employment, housing, health care, political power and education [25]. It has been shown to contribute to oral health inequalities [26], with empirical research demonstrating its role in Indigenous oral health inequalities [27,28]. 

Shiffman and Smith [29] developed the Political Power Framework, which comprises four categories: (i) the power of ideas; (ii) the power of the issue; (iii) the power of the actors; and (iv) the power of the political context. Each of these is considered below. 

In the setting of international Indigenous oral health inequalities, the “power of ideas” can be seen as the ways in which those involved with the issue understand and portray it. While many Indigenous leaders recognize the importance of reducing inequalities in general health and wellbeing, the voice for reducing inequalities specifically as they relate to oral health is small. This is likely influenced by the low numbers of Indigenous personnel in the oral health workforce internationally (relative to the general health workforce) [30]. 

The “power of the issue” relates to a lack of credible indicators in many countries in which Indigenous populations reside. These include measures that clearly indicate the oral health deficits between Indigenous and non-Indigenous groups that can be used to monitor progress, the size of the burden relative to other morbidity/mortality indicators and an inability to develop effective, realistic, culturally safe and cost-effective interventions. Current measures used to capture oral health inequalities—such as the decayed, missing and filled teeth (DMFT) index, slope index of inequality or relative index of inequality—do not sufficiently capture the inequalities arising from colonial influences that have resulted in the sustained loss of lands, identity, languages and the control to live life in a traditional, cultural way that is meaningful to many Indigenous groups.

The “power of the actors” relates to the strength of the individuals and organizations concerned with the issue which, as outlined above, is often small and under-resourced. This includes the degree of cohesion among the network of individuals and organizations that are centrally involved with the issue at the global level, leadership by individuals capable of uniting the policy community and who are acknowledged as strong champions for the cause, the effectiveness of organizations or coordinating mechanisms with a mandate to lead the initiative, and the extent to which grassroots organizations have been successful in pressuring international and national political authorities to address the issue. On all of these fronts, addressing Indigenous oral health inequalities at an international level has failed [31]. 

The “power of the political context” pertains to the environments in which actors operate. This includes the ability to identify political moments when global conditions align favorably for an issue, presenting opportunities for advocates to influence decision makers and the degree to which norms and institutions operating in a sector provide a platform for effective collective action. At an international level, this has been very difficult to operationalize with respect to Indigenous oral health inequalities. While many governments and health research funding agencies do identify the importance of reducing Indigenous health inequalities (for example, Australia’s National Health and Medical Research Council commits at least five percent of its medical research endowment account to Indigenous health [32]), this almost never extends to Indigenous oral health. The Indian Health Service (which provides dental services to American Indians and Alaskan Natives) recently estimated that it had 100 dentist vacancies, with a reduction in program funding predicted to limit its ability to fill the positions [33]. Recent estimates suggest that around 70 percent of Navajo children (the largest federally recognized tribe in the United States) have untreated dental decay [34], with fewer dental providers in the IHS substantially compromising their access to dental care. The concept of Indigenous self-determination and sovereignty is crucial when considering the power of the political context; in New Zealand, this manifests as tino rangatiratanga (“absolute sovereignty” as defined in the Māori version of the Treaty of Waitangi [35]) and mana motuhake (Māori self-rule and self-determination). What is also important is a deep understanding of Indigenous people’s disproportionate shouldering of the social and economic burden of neoliberalism (the dominant economic and philosophical model underpinning the operation of most transnational corporations and governments, involving markets that are private and competitive; social services and infrastructure receiving reduced public expenditure; and economic activity and freedom of choice facilitated by deregulation). This has, over time, resulted in Indigenous populations having poorer health over and above any issues of differential access to services [36,37].

Some countries have implemented policies and strategies that seek to improve Indigenous oral health. For example, Universities in New Zealand and Australia that provide training for oral health practitioners have designated enrolment quotas specifically for Maori/Aboriginal/Torres Strait Islander students. Canada is proposing to increase taxes on sugar-sweetened beverages (SSBs); while not directly targeting Aboriginal Canadians, this is likely to have an impact, given the high rates of SSB consumption among this group. In Brazil, a subsystem of Indigenous Health Care was incorporated into the Unified Health System. It was created to guarantee Indigenous people’s access to comprehensive health care (including dental health care), and takes into account the barriers that make this population more vulnerable to poor health outcomes. 

Crucial strategies for improving Indigenous oral health inequalities at a global level are intervening at key leverage points, acting simultaneously on multiple subsystems, and counteracting the social determinants of health. There are a number of examples of how this might be realized. First, the establishment of an international Indigenous oral health consortium and leadership team to drive both advocacy for improvements in Indigenous oral health and to provide support for emerging Indigenous oral health personnel and researchers. Second, it would be useful to establish specific Indigenous oral health special interest groups in international dental organizations, including the International Association of Dental Research and FDI World Dental Federation. Third, the World Health Organization should recognize the specific needs and rights of Indigenous populations in terms of oral health. Fourth, the United Nations should be urged to include oral health in its programs specifically targeting Indigenous populations. Finally, there is a strong case for supporting the development of conferences and general meetings on Indigenous oral health inequalities, and encouraging participation and leadership by Indigenous groups from around the world, with a view to increasing political advocacy at a global level.

To facilitate reducing Indigenous oral health inequalities at a global level, it is first important to have a solid understanding of the magnitude of inequalities that exist for Indigenous populations residing in non-Western countries (who do not tend to have regular or representative national surveys of oral health). Indigenous people live in nearly 90 countries, yet Indigenous oral health information tends to come only from Australia, New Zealand, Canada, the United States and Brazil. Structural discrimination that is present in almost all countries can only be realistically tackled when the socio-political context that allows such discrimination is tackled; this will have positive impacts on a wide range of health states, not just oral health. 

Article 24 of the United Nations Declaration on the Rights of Indigenous Peoples [38] states that “Indigenous individuals have an equal right to the enjoyment of the highest attainable standard of physical and mental health”. This includes oral health. However, some of the most severe Indigenous health inequalities are found in respect to oral health. This is largely due to past policies of forced assimilation and cultural annihilation, as well as contemporary social determinants of health and systemic racism. Shiffman and Smith’s Political Power Framework provides a helpful schema through which to analyze the power structures that contribute to the sustenance of these inequalities, including the power of ideas, the power of the issue, the power of the actors and the power of the political context. Supporting and developing structures that foster collaboration and leadership for emerging Indigenous oral health leaders may enable a momentum to develop that facilitates Indigenous oral health inequalities becoming a recognized issue among the political and societal organizations that have the power to enable change.

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
