# Peer review of "Indigenous Oral Health Inequalities at an International Level: A Commentary"

_ijerph, 2020, doi:10.3390/ijerph17113958_

Round 1
Reviewer 1 Report
Dear authors,
This commentary descrived the overview of indigenous oral health inequalities. It summarized the history and explained the potential reason why indigenous oral health inequalities persist. Some points are requiring further explanation, which would make the commentary more informative.
#1. It would be persuasive by adding numerical summaries of indigenous oral health inequalities (e.g., average DMFT, the number of tooth loss, access to dental care) in some countries in the first paragraph.
#2. Shiffman and Smith’s Political Power Framework explains the cause of indigenous oral health inequalities well. In the power of the issue category, a lack of credible indicators is argued as one of the problems. It would be informative for readers to add a deeper explanation of why the existing indicators of oral health, such as DMFT etc., and indicators of health inequality, such as slope index of inequality or relative index of inequality, are not sufficient in this context.
#3. There would be several policies and interventions aiming to reduce indigenous oral health inequality in some countries. Summarizing them, for example, adding one paragraph about it, would also be informative.
Author Response
#1. It would be persuasive by adding numerical summaries of indigenous oral health inequalities (e.g., average DMFT, the number of tooth loss, access to dental care) in some countries in the first paragraph. AUTHOR RESPONSE: Numerical summaries of prevalence of untreated dental caries between Indigenous and non-Indigenous populations from National Oral Health Surveys for Australia, Canada and New Zealand now provided on Page 1, Paragraph 1. Also provided are pooled estimates of dmft/DMFT for Indigenous groups in four South American countries.
#2. Shiffman and Smith’s Political Power Framework explains the cause of indigenous oral health inequalities well. In the power of the issue category, a lack of credible indicators is argued as one of the problems. It would be informative for readers to add a deeper explanation of why the existing indicators of oral health, such as DMFT etc., and indicators of health inequality, such as slope index of inequality or relative index of inequality, are not sufficient in this context. AUTHOR RESPONSE: Limitations of existing measures of oral health inequalities for Indigenous populations now provided on Page 3, Paragraph 1.
#3. There would be several policies and interventions aiming to reduce indigenous oral health inequality in some countries. Summarizing them, for example, adding one paragraph about it, would also be informative. AUTHOR RESPONSE: An additional paragraph has been added (Page 3, Paragraph 4) outlining some of the strategies being currently adopted by some countries to ameliorate Indigenous oral health inequalities.
Reviewer 2 Report
This is a timely commentary on a forgotten area of dental public health. The authors are praised for moving the agenda of indigenous oral health inequalities. I only have a couple of suggestions:
One is to include some hard (epidemiological) data to describe the magnitude of indigenous oral health inequalities. How large are disparities in oral health (both in absolute and relative terms) between the main and indigenous populations in (let's say Western) countries? A table summarising these differences will strengthen the arguments presented.
The other is to add a section at the end outlining gaps in knowledge, future areas for research and potential interventions to address these disparities. For instance, what kind of information is currently needed to advocate for indigenous oral health at a global scale? Surely, most data come from Western countries, but if the authors are thinking globally, what other indigenous populations around the world we don't have information on? The authors talk about structural discrimination but it would be good to know of any policies that can reduce such negative influences (either for oral health alone or multiple health conditions).
Author Response
- One is to include some hard (epidemiological) data to describe the magnitude of indigenous oral health inequalities. How large are disparities in oral health (both in absolute and relative terms) between the main and indigenous populations in (let's say Western) countries? A table summarising these differences will strengthen the arguments presented. AUTHOR RESPONSE: This data now provided in Page 1, Paragraph 1 in response to Reviewer 1's requests.
- The other is to add a section at the end outlining gaps in knowledge, future areas for research and potential interventions to address these disparities. For instance, what kind of information is currently needed to advocate for indigenous oral health at a global scale? Surely, most data come from Western countries, but if the authors are thinking globally, what other indigenous populations around the world we don't have information on? The authors talk about structural discrimination but it would be good to know of any policies that can reduce such negative influences (either for oral health alone or multiple health conditions). AUTHOR RESPONSE: A paragraph added (Page 4, Paragraph 2) that outlines gaps in knowledge, future ideas for research and potential interventions.
Reviewer 3 Report
page 2 line 58: approx. 900,000, not 100,000,000
page 2 line 58: persons, not peoples
page 5 lines 202 et seq.: correct source numbering
Author Response
- page 2 line 58: approx. 900,000, not 100,000,000. AUTHOR RESPONSE: Sorry, we could not find the figure of 100,000,000
- page 2 line 58: persons, not peoples. AUTHOR RESPONSE: Changed ‘peoples’ to ‘persons’
- page 5 lines 202 et seq.: correct source numbering. AUTHOR RESPONSE: All reference/reference numbering is correct.
Round 2
Reviewer 2 Report
The authors have addressed all of my comments. No further changes needed.